# Constructed Wetland Coupled Microbial Fuel Cell: A Clean Technology for Sustainable Treatment of Wastewater and Bioelectricity Generation

**Shiwangi Kesarwani** [1,†]**, Diksha Panwar** [1,†]**, Joyabrata Mal** [1]**, Nirakar Pradhan** [2] **and Radha Rani** [1,*]

[1] Department of Biotechnology, Motilal Nehru National Institute of Technology Allahabad, Teliyarganj, Prayagraj 211004, Uttar Pradesh, India
[2] Department of Biology, Faculty of Science, Hong Kong Baptist University, Kowloon Tong, Hong Kong
[*] Correspondence: radharani@mnnit.ac.in or raadharaani1982@gmail.com
[†] These authors contributed equally to this work.

**Abstract:** The availability of clean water and the depletion of non-renewable resources provide challenges to modern society. The widespread use of conventional wastewater treatment necessitates significant financial and energy expenditure. Constructed Wetland Microbial Fuel Cells (CW-MFCs), a more recent alternative technology that incorporates a Microbial Fuel Cell (MFC) inside a Constructed Wetland (CW), can alleviate these problems. By utilizing a CW's inherent redox gradient, MFC can produce electricity while also improving a CW's capacity for wastewater treatment. Electroactive bacteria in the anaerobic zone oxidize the organic contaminants in the wastewater, releasing electrons and protons in the process. Through an external circuit, these electrons travel to the cathode and produce electricity. Researchers have demonstrated the potential of CW-MFC technology in harnessing bio-electricity from wastewater while achieving pollutant removal at the lab and pilot scales, using both domestic and industrial wastewater. However, several limitations, such as inadequate removal of nitrogen, phosphates, and toxic organic/inorganic pollutants, limits its applicability on a large scale. In addition, the whole system must be well optimized to achieve effective wastewater treatment along with energy, as the ecosystem of the CW-MFC is large, and has diverse biotic and abiotic components which interact with each other in a dynamic manner. Therefore, by modifying important components and optimizing various influencing factors, the performance of this hybrid system in terms of wastewater treatment and power generation can be improved, making CW-MFCs a cost-effective, cleaner, and more sustainable approach for wastewater treatment that can be used in real-world applications in the future.

**Keywords:** constructed wetland; microbial fuel cell; wastewater treatment; bioelectricity generation; sustainable energy generation

## 1. Introduction

Environmental pollution and the lack of non-renewable energy resources due to a growing population and global warming are two of the biggest problems facing the Earth today. Owing to this, scientists today are focused on introducing cleaner technologies to manage wastewater and produce sustainable energy [1]. The water resource accounts for 71% of Earth's surface, but about 97% of this water is held by the ocean and cannot be utilized by humans for drinking, crop production, or other industrial use; thus, only 3% of the Earth's water is freshwater, of which only 0.5% is present in an available form [2].

India, which is an agriculture-based economy, has 4% of the world's freshwater reserves, of which 80% is used for agriculture. Out of the total 4000 billion cubic meters (BCM) of precipitation (approximately) received per year, about 48% recharges the surface and groundwater bodies. In India, total utilizable water resources have been estimated to be 1123 BCM, and about 688 BCM of water is used for irrigation, which may increase

to 1072 BCM by 2050. However, as a result of anthropogenic activities such as increased industrialization and excessive resource exploitation, surface water is becoming more and more contaminated (approximately 70% of India's water resources are now polluted), and groundwater is becoming more and more scarce. Due to the discharge of untreated domestic and industrial effluent into natural water bodies, levels of organic, inorganic, and microbial pollutants have increased, which has caused high levels of pollution in Indian rivers. According to the Central Pollution Control Board (2015), 61,754 million liters per day (MLD) of sewage are generated in India, while 37% of the wastewater can be treated, i.e., 22,963 MLD [3,4].

The factors influencing the water pollution are intensive agriculture, agricultural runoff, industrial effluent, industrial production, untreated urban wastewater runoff, and domestic sewage. According to the World Health Organization (WHO), half of India's morbidity is due to contaminated or polluted water [5]. In addition, with the ever-increasing population and industrial landscape, waste generation has accelerated significantly. Its management has also become more challenging, thus leading to significant damage to the ecosystem. Carbon (organic matter) is the major pollutant in the wastewater and treatment processes; to oxidize them to harmless by-products, such as carbon dioxide and water, requires a significant expenditure of energy and, thus, is not considered as a sustainable method [6]. Wastewater treatment plants (WWTPs) are frequently designed to satisfy certain effluent water quality requirements without sufficient consideration given to energy requirements, which represent about 25–40% of the operational cost of WWTPs. To purify the wastewater, conventional wastewater treatment methods are used, but they often consume high amounts of energy, and, thus, are significantly cost intensive. One good alternative is to extract energy from the wastewater itself during the treatment. Wastewater has high potential energy, and anaerobic digestion of it is already known to produce biogas. Integration of microbial fuel cells (MFCs) into the treatment process is an alternate way of producing sustainable energy, and a way to accomplish this is the coupling of MFCs with the natural redox gradient of Constructed Wetlands (CWs). A CW allows for the removal of pollutants and organic matter from wastewater by biological processes, such as filtration, adsorption, accumulation by plants, degradation, and transformation, by microorganisms [6]. The matrix of the wetland, which can consist of soil/charcoal/sand/pebbles, etc., also helps with the physical separation of suspended solids in wastewater, to some extent. Wetlands generate aerobic and anaerobic zones that can be utilized as a means for MFCs to harvest electricity. Aconstructed wetland microbial fuel cell (CW-MFC) is a hybrid of wetland and microbial fuel cell models with the potential to treat wastewater and, simultaneously, generate clean electricity. The generation of electricity takes place due to the chemical reaction, substrate accumulation, and bio-interaction caused by exoelectrogenic microbes thriving on organic matter in waste in the anaerobic zone of the wetland. A natural or constructed wetland itself contains both an aerobic and an anaerobic zone, which generates a redox gradient. This gradient is utilized to divert and allow the flow of electrons (produced by anaerobic microbial metabolism) into the desired route, thus leading to the flow of electricity, as in an MFC.

The integrated approach CW-MFC has been demonstrated to be effective not only for the removal of COD, BOD, total nitrogen, and phosphorus, but also harmful pollutants such as reactive dyes, pharmaceuticals, pesticides, hydrocarbons, heavy metals, and many more, along with energy output in the form of electric currents [7–9].

Therefore, constructed wetlands coupled with microbial fuel cells offer an effective solution for sustainable, economic, and energy-efficient technology for the treatment of wastewater with simultaneous bio-electricity production. The paper comprehensively reviews the reported applications of CW-MFC for wastewater treatment and energy generation, with prospective paradigms, limitations, and future perspectives.

**2. Critical Pollutants in Industrial Wastewater and Their Ecological Effects**

Some of the important pollutants commonly associated with industrial effluents and their ecological effects have been briefly discussed as follows.

*2.1. Heavy Metals*

Heavy metals such as Lead (Pb), Mercury (Hg), Cadmium (Cd), Chromium (Cr), Arsenic (As), Zinc (Zn), Cobalt (Co), Copper (Cu), and Iron (Fe) are predominantly present in wastewater generated by the mining, smelting, electroplating, tanning, and refining industries. Metals such as mercury are highly toxic, and can enter into the food chain. As a result of this, biomagnifications take place that can affect the aquatic ecosystem and human life. The exposure of a small amount of toxic heavy metals can lead to damage to the nervous system and kidneys [10].

*2.2. Radionuclides*

Major sources of radioactive pollution in wastewater are the emissions from nuclear power plants, the testing of nuclear weapons, nuclear waste disposal, and mining of uranium. Major radionuclides in polluted water include radon (Rd), uranium (U), thorium (Th) etc. [10]. Radioactivity can result in the formation of free radicals upon interaction with biomolecules, which can damage DNA and lead to cancer, birth defects, and even death [11].

*2.3. Pharmaceuticals*

The effluents generated from the pharmaceutical industries consist of drugs and drug metabolites (such as antibiotics, analgesics, anti-inflammatory compounds, etc.) that can cause harmful effects in the environment, as well as in the organisms. Pharmaceutical wastewater generally has a low C:N ratio, high sulfate concentration, complex composition, and high mammalian/biological activity. Moreover, the presence of antibiotics in the aquatic system can result in active selection of resistant microorganisms [10].

*2.4. Dyes*

The textile industry consumes a large amount of dyes. Apart from the textile industry, several other industries participate in the production and consumption of these dyes. Such industries include cosmetics, paper, printing, plastic, etc. Azo dyes can cause allergies, genotoxicity, and mutagenicity in humans and animals [10].

*2.5. Pesticides and Endocrine Disruptors*

The consumption of pesticides is increasing in order to improve the quality and the productivity of the crops that can become real threats for the aquatic environment. The agricultural runoff contains residue of pesticides, which are able to affect the structure and function of invertebrates. Long-term and low-dose exposure is sufficient to disrupt the endocrine system, cause hormonal disturbance, and lead to infertility. These pesticides can affect the nervous system as well as induce diabetes, cognitive impairment, and sperm DNA damage. Bis-phenol A and alkyl phenols affect the development of mammary glands in animals [12].

*2.6. Hydrocarbon Compounds*

Hydrocarbon pollutants come from the by-products of the petroleum, pesticides, crude oil refining, and processing industries, which contain huge amounts of toxic aliphatic and aromatic compounds in their wastewater effluents. Phenolic compounds can be easily absorbed through the skin, and once inside the system, they can interact with the proteins and exert toxic effects [12]. Hydroquinone is a phenolic metabolite with many industrial applications, including in paints, motor fuels, varnishing oils, etc., and human exposure to it can have toxic effects and is known to damage DNA [13].

## 2.7. Microplastics and Other Micropollutants

Microplastics are 1–1000 μm in size, and are the major cause of pollution in the aquatic ecosystem. They cannot be removed by conventional wastewater treatment. Sources of primary microplastics include plastic pellets, 3D printing powders, and industrial abrasives, while secondary microplastics originate from larger plastics, and include pollutants such as dust from vehicle tires, fragments of textiles, etc. These microplastics, owing to their high sorption, can carry micropollutants such as pesticides, pharmaceuticals, metals, etc. [14].

## 2.8. Persistent Organic Pollutants (POPs)

These are anthropogenic compounds produced and released in the environment either intentionally or unintentionally. These pollutants, once released, can be carried long distances by wind and streams, contaminating air and water bodies. POPs such as dichlorodiphenyltrichloroethane and polychlorinated biphenyls, once consumed by marine animals, can accumulate in fatty tissues and bioaccumulate through the food chain, leading to high exposure for upper trophic level predators. Some POPs have structural similarities, with hormones disrupting the endocrine functions. High exposure to DDT insecticide can cause vomiting and seizures. High exposure to fungicide hexachlorobenzene, which is used for treating seed grains, can adversely affect the nervous system [15].

The key pollutants in industrial wastewater, as well as their origins and impact, have been summarized in the Table 1.

**Table 1.** Summary of pollutants in water, their origins, and their impact.

| Pollutant | Origin | Impact |
|---|---|---|
| Heavy metals (mercury, chromium, arsenic, lead etc.) | Electronic and electroplating plant<br>Food and beverage processing industry<br>Rubber processing industry | Highly toxic, can accumulate through the food chain. Mercury is highly toxic in the nervous system. Lead affects mental capabilities in children |
| Radionuclides | Naturally from Earth's crust<br>Nuclear power plants<br>Nuclear weapon testing and manufacturing | Damages DNA<br>Cancer |
| Xenobiotics | Pharmaceutical industries | Emerging antibiotic resistance in pathogens<br>Increases BOD, COD in water |
| Dyes (azo dye, Sulfur dyes) | Paper, printing, textile, and cosmetic industries | Impairs the photosynthetic process<br>Accumulates in the food chain and are recalcitrant<br>Mutagenic and carcinogenic |
| Pesticides and herbicides | Agricultural run-off<br>Mill waste | Allergenicity<br>Affect neuro-endocrine system |
| Microplastics | 3D printing powders<br>Industrial abrasives<br>Tyre manufacturing | Can adsorb micropollutants<br>Cannot be removed by conventional wastewater treatment |
| Persistent organic pollutants (POPs) | Pesticide industries<br>By-products of industrial processes and combustion | Global circulation, accumulate in food web, noxious to living creatures, may act as endocrine disruptors |

## 3. Conventional Wastewater Technologies

There are three steps in wastewater treatment to remove organic/inorganic and toxic substances, as well as to kill pathogenic microbes in the wastewater.

## 3.1. Primary Treatment

The removal of large particles involves two steps.

Preliminary treatment: This consists of screening through the grit chamber and skimming tank to remove large particles and debris.

Screening: The aim of screening is to protect mechanical equipment and to prevent the clogging of valves and other accessories from the wastewater treatment plant by removing large and suspended particles, such as plastic, wood, paper, and cloth.

Grit chamber: This is used to remove sand metal fragments and broken glass from wastewater in order to protect pumps and other mechanical equipment from the wastewater treatment plant. Sometimes, an aerated grit chamber is used, whereby air bubbles are injected into the wastewater to strip off the organic material from the surfaces of the inert grit and to maintain the proper flow rate.

Skimming tanks: This is used to remove floating matters, e.g., oil, wax, fat, grease, and soap. These floating matters can adversely affect the activated sludge performance.

Sedimentation process: this process involves chemical precipitation in primary settling tanks to remove organic solids and suspended particles in the form of sludge.

Primary settling tank: The principle of the settling tank is gravitational separation. It is used to remove organic matter and suspended particles from the wastewater. These organic matters increase the oxygen demand and reduce disinfection efficiency in the subsequent treatment process. This step also includes a physical and chemical sedimentation process to remove fine particles.

### 3.2. Secondary Treatment

To remove dissolved and collided pollutants, as well as soluble materials that require oxygen for decay, suitable types of microbes are added to the wastewater in the presence of oxygen. As a result of this, biochemical oxygen demand (BOD) decreases.

Aerobic attached growth system: The aerobic bacteria begin oxidizing the organic matter, as a result of which a bacterial film or slime layer forms on the wastewater surface; consequently, it will settle down.

Trickling filters: This is a cylindrical tank filled with coke, gravel, ceramic, polyurethane foam, etc. These substances provide a large surface area for the biofilm formation.

Aerobic suspended growth system: aerobic microbes are suspended in the liquid medium with continuous mixing, and these microbes convert the organic matter into gases.

Activated sludge process: Effluent from the primary clarifier and microbes from the secondary clarifier or activated sludge are collected into the aeration tank. In the aeration tank, aerobic conditions are maintained for the oxidation of organic matter to remove impurities. After agitation, the treated water is sent to the secondary clarifier.

Aerated lagoon: Mechanical aerators are fixed in the lagoons to oxidize the organic matter by using suspended aerobes. There are two types: facultative aerated lagoons and anaerobic lagoons. The aerobic degradation takes place in the upper layer, while the solid matter settles down at the bottom layer, where anaerobic degradation subsequently takes place in the facultative aerated lagoon.

Oxidation pond: In this system, the bacteria metabolize the organic matter. Due to this, various inorganic nutrients, such as nitrogen, phosphorus, and carbon dioxide, are released into the wastewater. These compounds are utilized by algae in the presence of sunlight, and they produce oxygen, which is taken up by bacteria. The BOD is decreased by closing the cycle.

### 3.3. Tertiary Treatment

This is used to remove remaining undesirable substances in the treated water, as well as to remove phosphorus, nitrogen, biodegradable organic matter, heavy metals, pathogenic bacteria, and viruses [16].

### 3.4. Limitations of Conventional Waste Water Treatment Method

In conventional wastewater treatment, microbial reactions occur, but chemicals are also required. Without them, treatment cannot efficiently remove high organic impurities, and takes a long time. High sludge and bad odor are produced after aerobic treatment. Capital investment, energy, and land requirements are high.

On the contrary, CW-MFC is a hybrid technology with aims for both wastewater treatment and energy recovery. It has an advantage over conventional biological treatment, as it is effective for high organic pollutants, takes less time, no sludge is produced after treatment, and there is no need for post-treatment. Moreover, it is eco-friendly, aesthetically appealing (due to vegetation), and has the potential for eco-restoration when applied at large scale. Thus, it is a sustainable and green approach to treatment [17].

## 4. Constructed Wetland

Constructed wetlands are man-made or artificial wetlands that mimic natural wetland, and they are designed to treat wastewater coming from domestic sources or industries such

as petroleum refineries, the pulp and paper industries, textile industries, sugar factories, sewage water, greywater, and agricultural runoff under the controlled conditions. It is made up of a filter bed (sand, soil/or gravels), naturally occurring microbes, media, and aquatic plants that can survive in the wetland ecosystem. Straw and compost are also used for the removal of metal content present in the wastewater. Constructed wetlands play an important role in the processing, recycling, purification, and storage of water. Constructed wetland wastewater treatment does not require energy input, thus decreasing the cost of maintenance and operations. These also provide a multipurpose ecosystem, offering flood control and carbon sequestration.

In general, the wastewater effluent, following primary treatment, is further subjected to secondary treatment by CW, followed by tertiary treatment. Constructed wetlands are effective in removing organic and suspended solids, while the efficiency of nitrogen and phosphorous removal still remains low, unless using a tool specially designed to do so [18].

### 4.1. Components of a Constructed Wetland System

The primary components of the constructed wetland system include a lower impermeable layer, a layered gravel/sand/support matrix, an upper aerobic zone, a lower anaerobic zone, and plants.

Impermeable layer: This layer prevents the infiltration and reduces the negative effect of waste on the aquifers.

Gravel layer: This is a supporting layer, and provides nutrients to the vegetation zone and root zone. The vegetation zone consists of plant material that facilitates phytoremediation and improves the water quality. Additionally, the root zone helps with the bioremediation and denitrification.

Aerobic and anaerobic systems: In the aerobic zone, plants grow, and aerobic bacteria for the degradation of organic and inorganic content present in the water are found. In the anaerobic zone, anaerobic bacteria are found to degrade the contaminants [18].

### 4.2. Constructed Wetland can Be of Three Types According to Flow Regimes

Surface Flow Constructed Wetland (SF-CW): Wastewater flows above a shallow substrate of soil or another medium supporting the roots of halophytes (Figure 1). The surface layer of this system is aerobic, while the deeper layers of wastewater and soil are anaerobic. This system is usually used to treat mine drainage and agricultural runoff. Their operating and capital costs are low, but they have a lower efficiency with regard to removing contaminants [19].

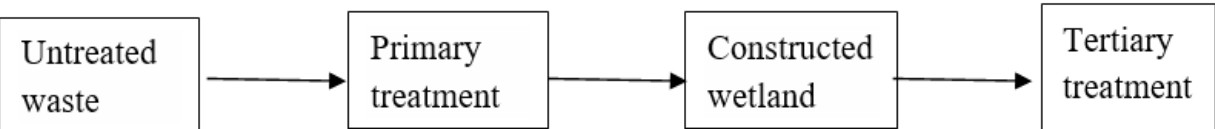

**Figure 1.** Constructed wetlands in a standard sewage treatment system.

Vertical Subsurface Flow-Constructed Wetland (VSSF-CW): Wetland with subsurface flow (SSF) consist of a sealed basin with a porous substrate of rock and gravel. In VSSF-CW, wastewater is fed in large batches and percolates down through the porous substrate, which allows the diffusion of oxygen from air to the bed (Figure 2). This type of wetland is used to treat municipal and domestic wastewater. It effectively reduces BOD, COD, suspended solids, metals, nitrogen, and phosphorus from wastewater [18].

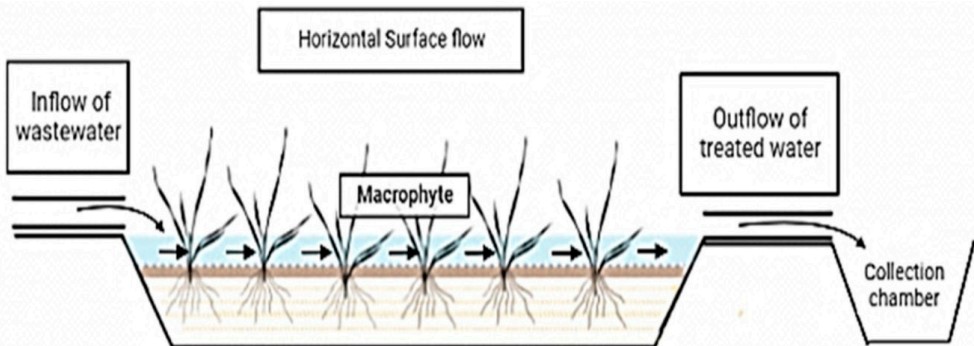

**Figure 2.** Surface flow-constructed wetland.

Horizontal Subsurface Flow-Constructed Wetland (HSSF-CW): This consists of an impermeable basin with a bed of gravels or rocks. Wastewater enters through the porous substrate under the bed surface, follows a horizontal path, and reaches the outlet (Figure 3). The aerobic zone is limited to the rhizosphere region, and the filtration beds remain anoxic. Organic matter is decomposed by microbial degradation in anaerobic conditions [18].

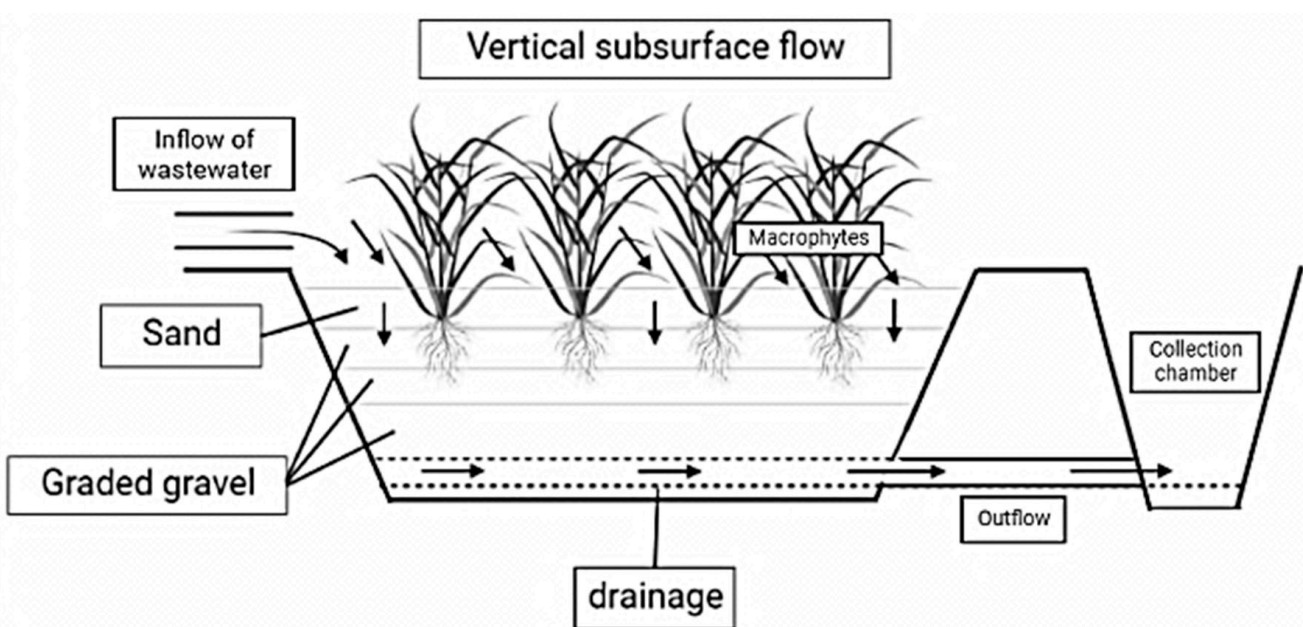

**Figure 3.** Vertical subsurface flow-constructed wetland.

## 5. Microbial Fuel Cell (Microbial Fuel Cell)

An MFC is a bio-electrochemical device that consists of cathode and anode chambers separated by a proton exchange membrane (Figure 4). In anaerobic conditions, oxidation of organic substrate in the anode chamber is performed by electrochemically active bacteria, which release electrons, protons, and $CO_2$. Protons cross PEM and reach the cathode, while electrons are conveyed to an external circuit and are used to generate electricity [20]. *Shewanella* and *Geobacter* are well-known exoelectrogenic bacteria that are able to perform electron transfer, either directly or indirectly, through biofilm [21].

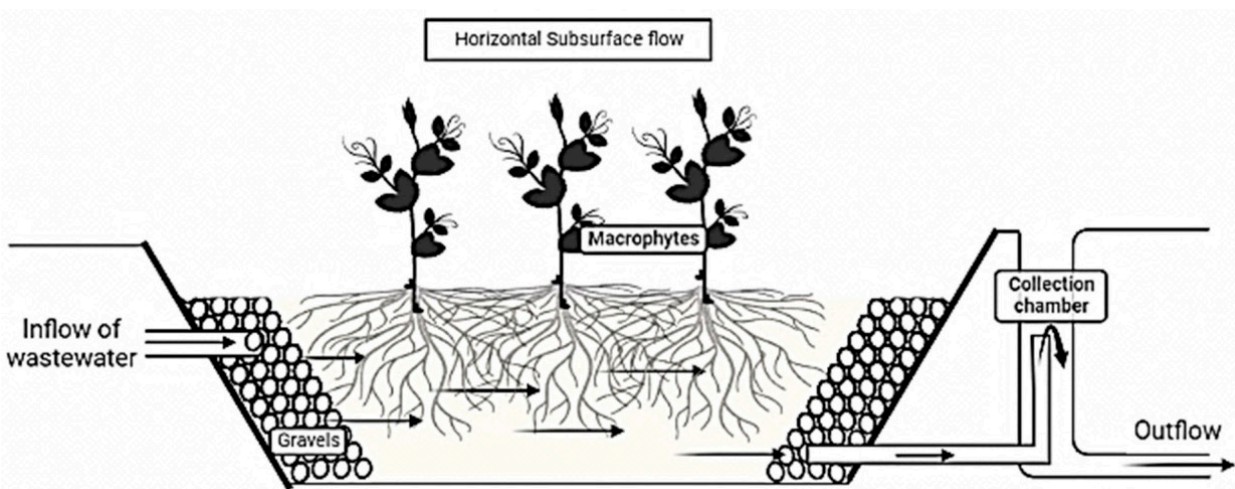

**Figure 4.** Horizontal subsurface flow-constructed wetland.

### 6. Constructed Wetland Integrated Microbial Fuel Cell

CW-MFC is a hybrid system in which an MFC is integrated into the constructed wetland [22]. The constructed wetland has an aerobic upper layer due to the diffusion of oxygen at the surface, gaseous exchange by plant roots, and growth of photosynthetic microalgae and other photosynthetic microbes. The deeper layers become anaerobic and decline in dissolved oxygen (DO) content. The absence of photosynthetic microbes leads to the existence of a naturally stratified redox gradient in these layers, as in the MFC, which is single-celled with two compartments [23]. In this integrated system, the conductive materials, i.e., the anode and cathode, are embedded in the upper layer (aerobic) and lower layer (anaerobic), respectively. Figure 5 shows a typical representation of a constructed wetland microbial fuel cell.

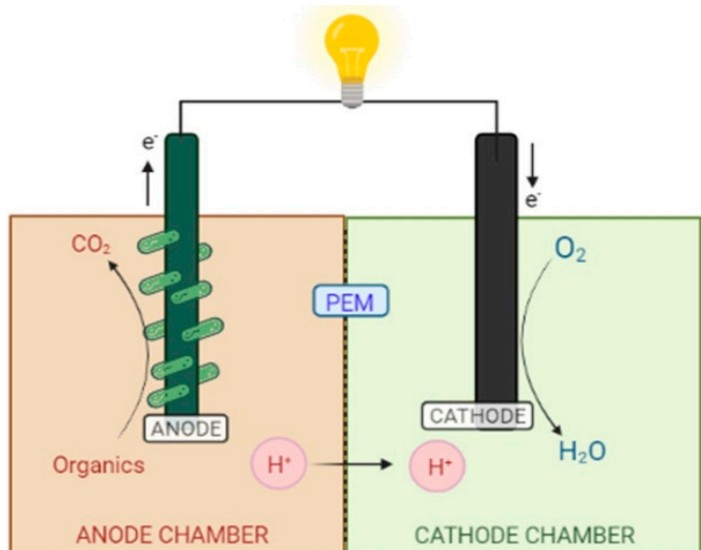

**Figure 5.** Microbial fuel cell (PEM = proton exchange membrane).

In the anodic chamber, which is located at the lower end of CW-MFC where the anode is deeply buried, anoxic conditions are maintained, which favors the existence of anaerobic microbial electricigens. These exoelectrogenic microbes utilize the organic components of wastewater anaerobically to produce electrons and protons. The generated electrons are collected by the anode and travel to the cathode via an external conductive wiring, thereby causing the flow of electric current. Removal of organic content and other pollutants can take place, both in the cathodic and anodic chambers, by microbial catabolism [1,24].

The cathodic chamber is located towards the upper side of the CW-MFC, where the cathode is placed in the root zone of the plants. This is an oxygenated chamber which lies in between the air–water interface, and aerobic bacteria are present in this zone. This compartment is planted with a macrophyte that promotes oxygen concentration through photosynthesis, and contributes to the reduction reaction. Oxidative microbial catabolism is favored due to high oxygen tension, and the microbial diversity is also high due to the presence of plant roots.

In a CW-MFC, apart from biological reactions, the substrate or the matrix of the system, along with the dense plant root system, participate in a physical separation/filtration of suspended impurities. This integration has the advantage of a triple interaction between the physical, chemical, and biological elements of the substrate, vegetation, and microbes for wastewater treatment and energy production [25,26].

### 6.1. Configuration of a CW-MFC

The aerobic and anaerobic compartments of a CW-MFC are separated by glass wool or different separators to create a sharp redox gradient for microbial reactions and electron transfer, which helps with the electricity generation and removal of contaminants. Some researchers, such as Yadav et al. [22], Villaseñor et al. [27], and Doherty et al. [1,24], constructed a CW-MFC by using glass wool or a bentonite layer as a separator. The position of the anode and cathode generated a redox gradient or a potential difference. These electrodes were connected together by copper/titanium wire with applied external resistance.

According to the flow patterns of CW, a CW-MFC can also be of three types viz. surface flow, horizontal subsurface flow (HSSF), and vertical subsurface flow (VSSF), as described in Section 4.2.

### 6.2. Principle Reactions Occurring in CW-MFC during the Operation

Plant roots release oxygen and root exudates (rhizodeposition), aerating the rhizosphere and promoting the microbial activity there. Ammonia-oxidizing bacteria and nitrite-oxidizing bacteria convert $NH_4^+$ (ammonium) into $NO_2^-$ (nitrite), and $NO_2^-$ (nitrite) into $NO_3^-$ (nitrate), respectively. Root exudates and organic compounds present in the wastewaterare anaerobically digested in the anode chamber, leading to the release ofelectrons, which are transferred to the cathode via an external circuit. Nitrite or $O_2$ act as final electron acceptors in the cathode [28] (Figure 6). The following reactions occur in CW-MFCs (Figure 7):

$$\text{Anode Chamber: } C_6H_{12}O_6 + 6H_2O \rightarrow 6CO_2 + 24H^+ + 24e^-$$

$$\text{Cathode Chamber: } 6O_2 + 24H^+ + 24e^- \rightarrow 12H_2O$$

$$\text{Overall Reaction: } C_6H_{12}O_6 + 6O_2 \rightarrow 6CO_2 + 6H_2O + \text{Electric Energy}$$

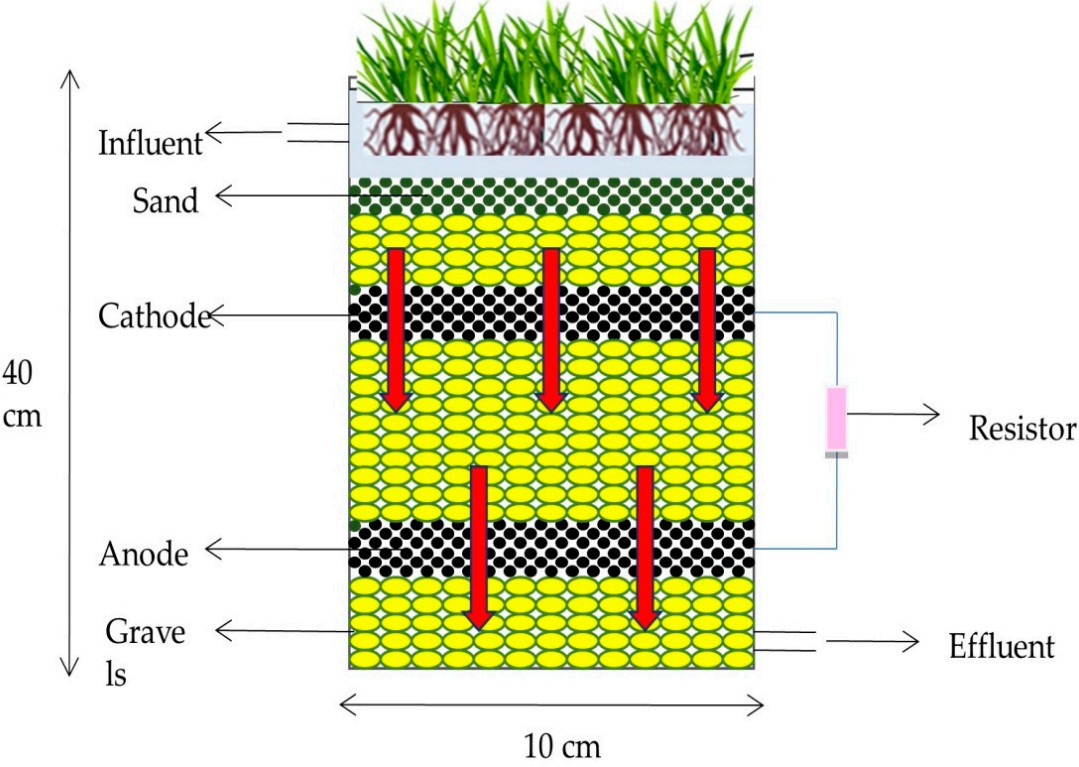

**Figure 6.** Constructed wetland microbial fuel cell (vertical subsurface flow).

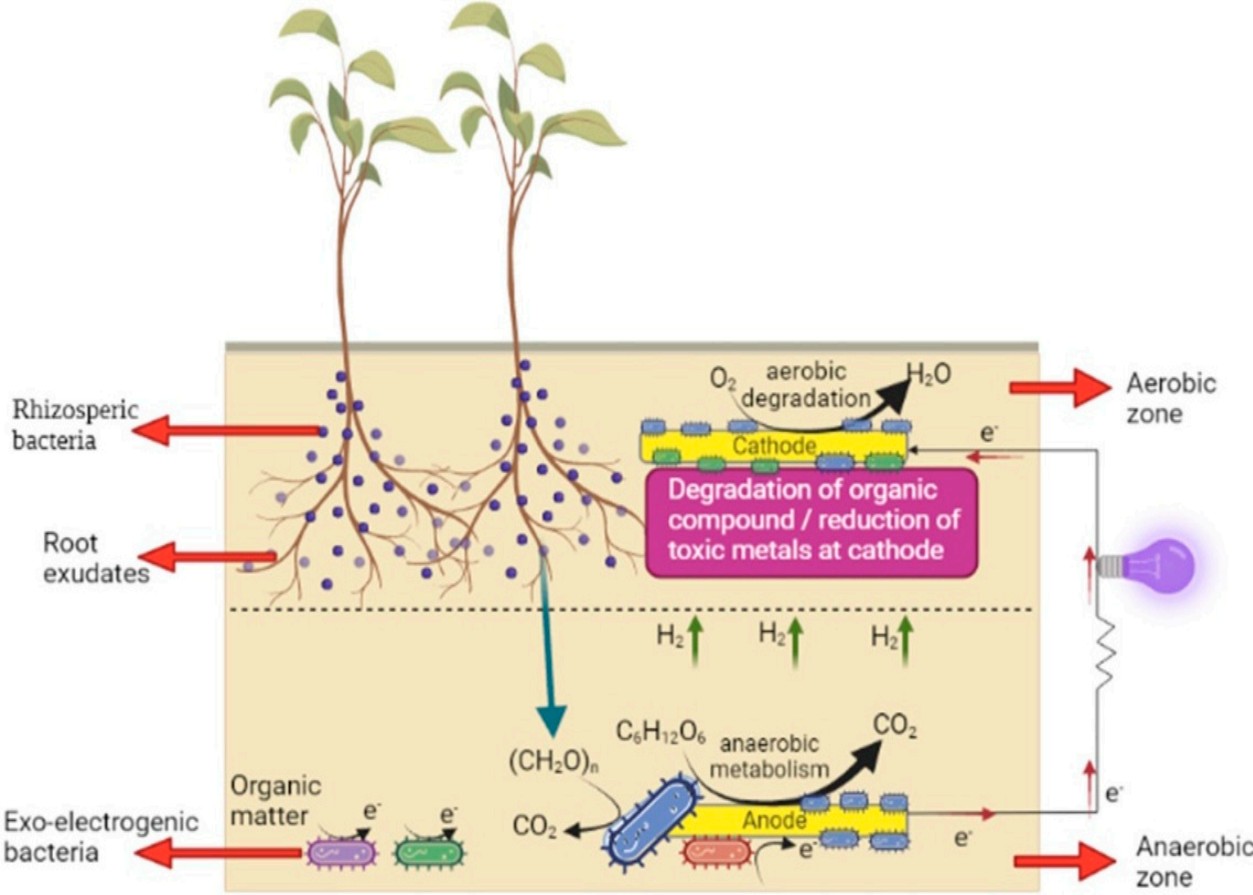

**Figure 7.** Principle reactions occurring in anodic and cathodic chambers of a CW-MFC.

### 6.3. Factors Affecting CW-MFC Performance

Wastewater treatment in a wetland occurs through various physiochemical and biological processes, such as filtration, sedimentation, adsorption, bioaccumulation, and denitrification by microbes in different sections of the wetland.

### 6.3.1. Wetland Macrophyte

The quantity and variety of the microbial population in CW-MFC, as well as their consequent impact on pollutant removal and bioelectricity generation, are often greatly enhanced by the presence of macrophytes [6]. Their roots offer a significant surface area for microbial development and adhesion, enabling the breakdown and uptake of contaminants from wastewater [29]. The main method for maintaining the system's effectiveness is to maintain a strong redox potential between the upper and lower areas of the CW-MFC. Wang et al. [30] conducted a study to evaluate the impact of aquatic macrophytes on the generation of bioelectricity, the degradation of contaminants, and the variety of the microbial communities in CW-MFCs. The 16S rRNA microbial analysis of their investigation revealed that the relative abundance of EABs (Firmicutes and β-Proteobacteria) in planted CW-MFCs was much higher than that in unplanted CW-MFCs. These results suggest that exoelectrogens around the anode material were significantly affected by CW-MFCs with aquatic plants. The resultant increase of 8.91 mW m$^{-2}$ in bioelectricity production in the planted CWMFC was caused by the higher relative abundance of EABs. On the other hand, photosynthetic CW-MFCs with smaller filler sizes achieved the maximum removal efficiency for COD and nitrate-nitrogen ($NO_3$-N) of 86.7% and 87.1%, respectively [6].

In the CW-MFC, oxidation-reduction (redox) events at the anode and cathode chambers, or electrochemical or biological reactions, are primarily what generate energy. In an experiment by Liu et al. [31], they contrasted CW-MFC systems with *Ipomoea aquatica* (water spinach) plants versus CW-MFC systems without plants. Their investigation clearly showed that the plants could increase the system's power generation by 142%. The system containing *Ipomoea aquatica* generated a maximum power density (MPD) of 12.42 mWm$^{-2}$, which was 142% higher than the 5.13 mW m$^{-2}$ produced by the system devoid of aquatic macrophytes [31]. The subsequent investigations byLiu et al. [32] evaluated the DO content of four CW-MFCs, including one that was unplanted and three that were planted with *Canna indica* (purple arrowroot), *Acorus calamus*, and *Ipomoea aquatica*. The unplanted CW-MFC had a DO concentration that was noticeably lower (1.95 mg/L) than the planted. When compared to the unplanted, the effluent from the CW-MFC that included *Canna indica* plants showed the greatest DO concentration (3.31 mg/L), which indirectly increased bioelectricity production [32].

Exudate secretion, however, also plays a significant part in the generation of bioelectricity in addition to the oxygenation of plant roots. Wetland plants emit organic matter as root exudates, which are oxidized by electroactive bacteria in the anode region to produce electrons for the generation of bioelectricity. The electrical output of the system will, therefore, be greatly influenced by macrophyte species with such a predisposition [33]. As the only endogenous substrate from their root exudate, Guadarrama-Pérez et al. [34] examined the impact of four different native species—*Aglaonema commutatum* (Chinese evergreen), *Epipremnumaureum* (Devil's ivy), *Dranacaenabraunni* (lucky bamboo), and *Philodendron cordatum* (heartleaf philodendron)—and the influence on the generation of bioelectricity. *Philodendron cordatum* produced a maximum of 20.6 mg/L of root exudates, and attained the highest power density, at 12.5 mW/m$^2$. Consequently, at ideal conditions for root exudates, the performance of CW-MFCs in terms of bioelectricity was enhanced. Different macrophyte species had varied effects, depending on the type of substrate they used, how they were transported, and how much exudate they emitted [34].

The macrophyte component has a variety of functions that help to effectively remove toxins from wastewater. Aquatic plants, for instance, have a significant direct or indirect role in the absorption, digestion, and storage of nutrients in their biomass during the bioaccumulation and phytoremediation processes. For the purpose of increasing their

biomass, they sequester soluble inorganic nutrients. Therefore, choosing the macrophyte with high efficiency in absorbing inorganic nutrients and converting them to organic biomass would maximize contaminant removal from wastewater [28,35,36].

In an experiment in South Africa by Oodally et al. [37], they compared three native South African species, *Phragmites australis* (Common reed), *Cyperus prolifer* (Paper reed), and *Wachendorfia thyrsiflora* (marsh butterfly lily), with an unplanted system as a control, to investigate the contribution of macrophytes employed in CW-MFCs. They found that *Cyperus prolifer* had a COD removal efficiency of 97%, which was higher than that of *Phragmites australis, Wachendorfia hyrsiflora*, and the unplanted, with removal efficiencies of 94%, 94%, and 90%, respectively. The orthophosphate removal efficiency of the *Cyperus prolifer* species was higher (98%) than that of *Wachendorfia thyrsiflora* (58%), *Phragmites australis* (81%), and the control experiment (72%). Comparatively, *Cyperus prolifer* was found to be the best native wetland plant for producing electricity and removing COD, ammonia, and phosphate [37].

### 6.3.2. Exoelectrogenic Microorganisms

These are bacteria that can generate electricity by oxidizing organic matter and transferring electrons to an acceptor outside the cell. These exoelectrogens or electroactive bacteria are rich in anaerobic sludge and anaerobic sediment of wastewater plants. They can be categorized into several groups based on the type of anaerobic respiration used by them (Table 2).

**Table 2.** Common groups of exo-electrogenic bacteria.

| Group | Example |
| --- | --- |
| Denitrifying bacteria | *Pseudomonas, Ochrobactrum* |
| Dissimilatory metal-reducing bacteria | *Geobacter, Shewanella, Geopsychrobacter, Geothrix* |
| Sulfate-reducing bacteria | *Desulfuromonas, Desulfolobus* |
| Fermentative | *Clostridium, Escherichia coli* |
| Purple non-sulfur, photosynthetic | *Rhodoferax ferrireducens* |
| Purple non-sulfur, non-photosynthetic | *Rhodopseudomonas palustris DX-1* |

Electroactive bacteria (EAB) or electrogens catalyze oxidation of the substrate in the anodic chamber, generating electrons, protons, and carbon dioxide. Protons are transferred to the cathode via PEM, while electrons are first transferred to the anode and then flow through the cathode via an external circuit, where they are used to convert $O_2$ to $H_2O$ in the cathode chamber. The circuit's electron flow produces a power output that may be measured and stored. It has been noted that the power output is influenced by the exoelectrogens' rate of substrate oxidation [38]. Richter et al. [39] used MFC technology to treat wastewater with exoelectrogen *Geobacter sulfurreducens*, a δ-Proteobacteria. They used gold electrodes and acetate and fumarate as an electron acceptor, and observed a current density of 3147 mA/m$^2$. In a similar study by Min et al. [40], a pure culture of *Geobacter metallireducens* resulted in a power output of only 40 mWm$^{-2}$ in an MFC using wastewater.

### 6.3.3. Microorganisms in Cathode

These are mostly aerobic bacteria that use oxygen present in the aerobic (cathodic) zone to degrade the pollutants present in wastewater. Microbes such as *Acidithiobacillus ferrooxidans* and *Thiobacillus ferrooxidans* as biocathodes can oxidize reduced metal oxides and $Fe^{+2}$ by indirect electron transfer. Denitrifying bacteria such as *Thiobacillus denitrificans*, *Micrococcus denitrificans* and *Pseudomonas* spp. can reduce nitrate to nitrogen. In a study by Sathiyanarayanan et al. [41], a biocathode with *Acidithiobacillus ferrooxidans* under constant polarization, $Fe^{+2}$ as an electron donor, and citrate as an iron chelator produced a maximum current density of −38.61 A/m$^2$ [41].

*Geobacter* sp can also be effectively used as a biocathode because of its ability to accept electrons from the cathode. *Geobacter metallireducens* can accept the electrons from cathode and reduce nitrate to nitrite. *Geobacter sulfurreducens,* similarly, can reduce fumarate to succinate [42]. In an experiment performed by Xafenias et al. [43], *Shewanella oneidensis* MR-1 in air-cathode MFCs showed an increased Cr(VI) reduction and a maximum current density of 32.5 mA/m$^2$. They also indicated the expression of riboflavin in electron transport. In a similar study by Freguia et al. [44], *Shewanella putrefaciens* and *Acinetobacter calcoaceticus* showed an increased rate of reduction of $O_2$ to $H_2O$ by using outer membrane-bound cytochromes and self-secreted PQQ.

### 6.3.4. Carbon Mass Balance in CW-MFC System

Natural wetlands are important ecosystems which contribute towards both the sequestration of carbon dioxide and the release of greenhouse gases, such as methane and carbon dioxide. Thus, constructed wetlands must also be considered from the point of view of carbon mass balance for their effective field applicability. In a CW, the anaerobic zone supports the growth of diverse anaerobes, including methanogens, sulfur reducers, and exoelectrogens, which can use the organic matter of wastewater for their respiration. In natural systems, the exoelectrogens and methanogens live in symbiotic relation, where the electrons released extracellularly by exoelectrogens are utilized by methanogens (or S reducing bacteria) for their respiration, with the release of methane [45]. However, in case of an integrated system, the CW-MFC's electron acceptor (anode) is embedded in the anaerobic zone, which uptakes electrons and transfers them to the cathode, where oxygen acts as the terminal acceptor and is reduced to water. In this reaction, i.e., $O_2{\rightarrow}H_2O$, the total energy gain is higher than the electron uptake by methanogens, $CO_2{\rightarrow}CH_4$ [46].

Hence, the presence of high-affinity electron acceptors in the form of anodes in a CW-MFC reduces the probability of electron uptake by methanogens, and, thus, methanogenesis, with simultaneous electricity generation. Therefore, CW-MFCs are more efficient systems in terms of carbon assimilation and sequestration, with limited release of greenhouse gases like methane [46].

### 6.3.5. Type of Wastewater/Substrate

Wastewater arising from different sources has different contaminants associated with it, which affect the performance of CW-MFC sinits treatment. Domestic wastewater arising from households is rich in organic load, which can affect the treatment efficiency and power density [27]. Wastewater from the textile industries contains recalcitrant dyes, which need to be decolorized during treatment. Oil refineries and the smelting industry cause the presence of heavy metals in effluent water, which need to be removed and recovered. There are different substrates that can be used in MFCs for energy production, such as glucose, maltose, acetate, and ethanol, which affect Coulombic efficiency, power density, and microbial composition [17]. According to Chae et al. [47], the performance of a microbial fuel cell is affected by the substrate. They employed four MFCs inoculated with anaerobic sludge and four distinct substrates in their investigation, namely acetate, butyrate, propionate, and glucose. According to their research, the Coulombic efficiency for acetate was 72.3%, butyrate was 43.3%, propionate was 36.6%, and the glucose-fed MFC was 15%. Because of the fermentative nature of glucose and the fact that non-electrons producing bacteria consumed it, the lowest Coulombic efficiency was observed for glucose. Additionally, they discussed how the substrate affected the microbial community in the anode chamber. They discovered that every substrate supplied to the MFC, with the exception of propionate, included β-Proteobacteria and Geobacter sp., but not γ-Proteobacteria. γ-Proteobacteria and Firmicutes predominated in the propionate-fed MFC. Additionally, they noted that the highest microbial diversity was found in glucose-fed MFCs, with rapid current production and no lag time [47].

### 6.3.6. Support Matrix/Media

Support matrix filtration media acts as a support matrix for living organisms and contaminants, and as a filtration medium that helps to eliminate the pollutant by filtration, trapping, adsorption, or biodegradation. Various physical (particle size, conductivity, porosity), chemical (surface charge), and biological (electron donor/acceptor) properties should be considered for a material to be used as a substrate. Zeolites are considered ideal due to their porosity, large surface area, adsorption stability, ability of riveting microorganisms, high nitrogen removal efficiency, low cost, and production of high power density [28].

### 6.3.7. Electrodes

The materials and positions of electrodes influence the performance of CW-MFCs by affecting the microbe–electrode interaction required for substrate oxidation, biofilm formation, and electron transfer [48]. The cathode is kept in the aerobic region and the anode is buried in anaerobic region, in order to utilize the natural redox gradient of the constructed wetland. An ideal electrode should have good conductivity, low internal resistance, and good mechanical strength; it should also be porous, biocompatible, and non-corrosive, with chemical stability and a large surface area [49]. Different electrodes that can be used include carbon-based electrodes such as graphite rods, carbon cloths, granulated activated carbon, carbon felt, metal-based stainless steel, titanium mesh, etc. In order to achieve a high power density, internal resistance needs to be lowered, which can be achieved by optimizing the electrode spacing and using appropriate electrode material [28]. Large electrode spacing results in increased internal resistance, causing ohmic losses and reduced maximum power densities [50].

Wang et al. [6] studied the influence of electrode spacing on performance of CW-MFCs in terms of power generation. They used three reactors, with electrode spacing of 10 cm, 20 cm, and 30 cm, respectively, and found that the electrode spacing of 10 cm yielded a maximum power density of 2.55 W/m$^3$, which was 30% and 50% higher than reactors with electrode spacing of 20 cm and 30 cm, respectively [6].

### 6.3.8. Hydraulic Retention/Resistance Time (HRT)

This is the average length of time that a soluble contaminant stays inside the reactor. HRT influences the contact between the substrate and the microorganisms. Increasing the HRT results in an increase in the efficiency of treatment by improving the pollutant removal rate and power generation. It is expressed by the following relationship [51].

$$HRT = \frac{\text{Reactor volume (V)}}{\text{Feeding amount } (\theta) * \text{No.of cycle per day (X)}}$$

Yang et al. [52] studied the performance of CW-MFCs at different HRTs of 6, 12, 18, 24, and 48 h, respectively. The study demonstrated that the internal resistance and the amount of time needed for the MFC-CW system to achieve a constant output voltage increase as HRT is extended. With the increase in HRT, the Coulomb efficiency slowly rises, but the power density eventually decreases. The reason, as analyzed in the study, was that when HRT is short, the shearing effect of water on the biofilm reduces the adhesion of non-conductive materials, boosts the rate of electron transfer, reduces internal resistance, and increases power density. In addition, as the HRT is raised, organic matter becomes trapped into the matrix, which allows the electrogenic bacteria to directly use the dissolved organic matter in the sewage and biodegrade it, thus increasing the Coulombic efficiency [52]. An HRT of 2–3 days is recommended for CW-MFCs [48]. As discussed in Section 7, Fang et al. [53] also demonstrated the influence of HRT on CW-MFC performance in terms of azo dye decolorization and electricity generation during wastewater treatment.

## 7. CW-MFC for Enhanced Wastewater Treatment and Electricity Generation

Integrating MFCs into constructed wetlands not only offers the advantage of treating wastewater economically and efficiently, but also the production of bioelectricity using the waste, thereby making the complete process energy-efficient and sustainable. Along with the removal of enhanced COD, BOD, nitrogen, and phosphate content in wastewater (comparative to the individual technology CW and MFC), MFCs also have significant potential for the removal of toxic pollutants. Use of this technology has been demonstrated for the sustainable treatment of different types of wastewater, including grey water, domestic wastewater, swine wastewater, food and beverage industry effluent, textile industry wastewater, heavy metal contaminated wastewater, and many others [17,30,54]. A detailed account on the use of CW-MFCs for wastewater treatment and electricity generation are presented in Table 3.

Xu et al. [55,56] studied the potential of CW-MFCs to treat municipal wastewater and generate electricity using microorganisms. The study reported a total nitrogen removal of 82.4%, an average COD removal of 82.3%, and an average phosphorus removal of 95%. The MFC had a maximum power density of 3714.08 mW/m$^2$ after 3 days of operation under continuous up-flow conditions using municipal wastewater, which was higher as compared to constructed wetlands [55,56]. In a study conducted by Srivastava et al. [57], COD removal efficiencies of 63–86% were reported in a CWMFC planted with *Canna indica* (purple arrowroot). This was achieved in batch mode, using synthetic wastewater, with different types of electrodes. They reported a maximum power density of 320.8 mW/m$^3$ and a maximum current density of 422.2 mA/m$^3$, using a granular graphite anode and platinum coated carbon cloth cathode, in the CW-MFC [58]. Yadav et al. [22] investigated the potential of CW-MFCs operating in batch mode to remove dye (methylene blue) from synthetic wastewater, along with electricity generation. They reported a 93.15% maximum dye removal rate when the initial dye concentration was 500 mg/L and the HRT was 96 h; however, at 1000 mg/L dye concentration, dye removal was 80%, with maximum power density and maximum current density of 15.73 mW/m$^2$ and 69.75 mA/m$^2$, respectively [22]. Fang et al. [58] studied the dye decolorization potential of an azo dye reactive brilliant red X-3B (ABRX3) with simultaneous electricity production in a CW-coupled MFC planted with *Ipomoea aquatica* (water spinach). They demonstrated a maximum decolorization rate of 91.24%, as well as a power density of 0.302 mW/m$^3$ in the CW-MFC onthe third day due to co-metabolism of glucose (180 mg/mL) and azo dye (150 mg/mL) when operated in continuous mode using X-3B simulated artificial wastewater [58]. The effects of HRT, COD (glucose and ABX3) concentration, and ABX3 proportion on the decolorization rate and electricity generation in CW-MFCs were also studied by Fang et al. [53]; they reported a maximum decolorization rate of 95.6% and a power density of 0.852 mW/m$^3$, with an HRT of 3 days, a COD concentration of 300 mg/mL, and 30% ABX3. However, there was a decline in the decolorization rate and power output upon further increasing HRT and dye concentration in wastewater, due to anodic polarization [53].

Yang et al. [59] demonstrated the influence of multiple factors on the performance optimization of CW-MFCs. They reported an increase in COD, nitrogen, and total phosphorous removal efficiencies by 6.06%, 3.7%, and 3.68% in CW-MFCs as compared to a traditional CWs. The maximum power density was 107.54 mW/m$^3$ when operated under optimal parameters, namely 200 mg/L initial COD, 24 h HRT, and 1000 Ω external resistance. Additionally, they indicated that pollutant removal efficiency and bioelectricity generation can be enhanced by gradual enrichment of electroactive bacteria and denitrifying bacteria [59]. Doherty et al. [50] studied the effect of electrode spacing and flow direction on the performance of CW-MFCs for swine wastewater treatment and electricity generation. They reported a maximum power density of 0.276 W/m$^3$ when electrode spacing was 0.4 m (with cathode at air-water interface), and operated with wastewater up-flow into the anode and down-flow into the cathode [50]. Similarly, Mu et al. [60] studied the effects of multiple factors, such as Cr(VI) and COD concentration, in wastewater, as well as HRT and electrode spacing in the unplanted CW-MFC, for Cr(VI) removal efficiency and electricity

generation. They reported a maximum Cr(VI) removal rate of 93.4% when the initial Cr(VI) in wastewater was 40 mg/mL, electrode spacing was greater than or equal to 10 cm, and HRT was 3 days. However, a maximum power density of 458.2 mW/m$^3$ was obtained at 60 mg/mL Cr(VI) and 500 mg/L COD, with Cr(VI) and COD removal efficiency values of 90.7% and 92.5% respectively [60].

The type of wetland vegetation also has an influence on the performance of CW-MFCs. Plant roots release oxygen, which affects the redox potential in the CWMFC. They also release exudates, which can act as a carbon source for denitrifying bacteria, thus enhancing nitrogen removal from the wastewater. Plant roots not only provide an attachment surface for microorganisms involved in degradation, but also provide adsorption and filtering effects for the removal of pollutants [28]. Saz et al. [61] demonstrated that planting CW-MFCs with *Typha angustifolia* led to higher treatment efficiency as compared to unplanted systems. They reported 85–88% COD, 95–97% NH$_4^+$, and 95–97% TP removal efficiency in a planted CW-MFC, with 7.47 $\pm$ 13.7 mW/m$^2$ maximum power density [61]. Oon et al. [62] studied the role of macrophytes and the effect of supplement aeration in up-flow CW-MFCs for the purpose of synthetic wastewater treatment and electricity generation. They reported 98% COD, 44% nitrate, and 84% ammonia removal efficiency, with a 184.75 $\pm$ 7.50 mW/m$^3$ maximum power density in the up-flow constructed wetland coupled with an MFC (UFCW-MFC) planted with *Elodea nuttalii*(Western waterweed) under artificial aeration of 600 mL/min [62]. Oodally et al. [37] investigated the performance of CW-MFCs using three indigenous South African wetland plants; they reported a maximum power density of 229 $\pm$ 52 mW/m$^3$, 97% COD removal, and 98% phosphorous removal efficiency using *Cyperus proliferas* (Paper reed) as the wetland plant [37]. *Villaseñor* et al. [27] used a horizontal subsurface flow constructed wetland (HSSF-CW) coupled an MFC and planted with *Phragmites australis* (common reed) to treat synthetic domestic wastewater with different organic loading rates. The study demonstrated that under a low organic loading rate of 13.9 g COD m$^{-2}$d$^{-1}$, t 80–100% COD removal efficiency could be observed, with a maximum power density of 0.15 mW [27]. Table 3 gives an overview of studies performed on CW-MFCs. For the purpose of comparing the effectiveness of a constructed wetland coupled with an MFC to a traditional constructed wetland for the simultaneous generation of electricity and the treatment of wastewater, several studies have been conducted. With the optimization of various parameters, this technology has the potential to contribute to a sustainable future.

**Table 3.** Application of CW-MFCs for wastewater treatment and power output.

| Dimension (h × d) (cm) | Vol (L) | Waste Water | Plants | COD Removal Efficiency (%) | TN Removal (%) | TP Removal (%) | Max. OCV (mv) | HRT (h) | Max. Power Density (mW/m$^2$) | Max. Current Density (mAm$^{-2}$) | References |
|---|---|---|---|---|---|---|---|---|---|---|---|
| 50 × 14.5 | 3.7 | swine wastwater | *Phragmites australis* | 76.5% | 49.7 | 65.9 | 495 | - | 9.35 | - | [63] |
| 115 × 47 | 150 | synthetic wastewater | *Phragmites australis* | 90–95% | - | - | 748 | 76.8 | 0.15 mW/m$^2$ | 1.1 mA/m$^2$ | [27] |
| 30 × 52.5 | 12.4 | synthetic wastwater with azo dye | *Ipomoea aquatica* | 86 | - | - | - | 72 | 0.302 W/m$^3$ | - | [58] |
| 10.5 × 62 | 5.4 | synthetic wastewater with methylene dye | *Canna indica* | 74.9 | - | - | - | 96 | 15.7 mW/m$^2$ | 69.75 | [22] |
| 30 × 50 | 35.3 | synthetic wastewater | *Ipomoea aquatica* | 94.8 | 90.8 | - | 530 | 48 | 12.42 | - | [31] |
| 25 × 45 | 6 | synthetic | *Cyperus* | 72 | 47 | 86 | 440 | 9 | 30 | 70 | [64] |
| 30 × 9 | 1.9 | synthetic | *Carex nigra* (Common Sedge) | 99.5 | 90 | - | 80 | 15 | - | 80 | [65] |

**Table 3.** *Cont.*

| Dimension (h × d) (cm) | Vol (L) | Waste Water | Plants | COD Removal Efficiency (%) | TN Removal (%) | TP Removal (%) | Max. OCV (mv) | HRT (h) | Max. Power Density (mW/m²) | Max. Current Density (mAm⁻²) | References |
|---|---|---|---|---|---|---|---|---|---|---|---|
| 0.7 × 0.17 | 8.1 | sswine wastewater | Phragmites australis | 93 | 85 | 98 | 280 | 24 | 383 | 856 | [66] |
| 18 × 75 | _ | synthetic | *Typha latifolia* | 100 | - | - | 421.7 | 24 | 6.12 | - | [67] |
| 20 × 55 | | municipal wastewater | *Phragmites australis* | 82.32 | 82.46 | 95.06 | 265 | 72 | 3714 | - | [55,56] |
| 30 × 50 | 35.3 | synthetic wastewater | *Phragmites australis* | 94.9 | - | - | 741 | 48 | 0.2 | - | [68] |

## 8. Economic Considerations

Economic assessments of small- to medium-scale constructed wetlands have been reported by several researchers, evaluating the suitability and whole life costing in comparison to other commonly used wastewater treatment technologies. Freeman et al. (2019) [69] compared the economics of a submerged aerated filter (SAF), a rotating biological contactor (RBC), and a saturated vertical flow (SVF) aerated wetland by whole life cost assessment for a small scale system (population equivalent, about 2000). They reported that both the capital and operating expenditures were comparatively low for the CW, followed by RBC and then SAF, over 40 years of operation. The study concluded that CW technology is an economic and effective technology to deliver long-term economic cost benefits in comparison to other commonly used wastewater treatment technologies. Teng et al. (2012) [70] evaluated the environmental and economic benefits of riparian-constructed wetlands (CWs) for the treatment of municipal wastewater, and reported that the total costs of the CWs were between USD 0.425 and 3.621 per kg of total BOD removed, while the cost of the centralized wastewater treatment plant was approximately USD 1.186 per kg of total BOD removed, thereby concluding that wetlands not only provide a considerable capacity for pollutant removal, but demonstrate additional benefits for recreation. Several other studies have also reported that the capital and operative costs involved in CWs are relatively lower than those involved in other methods used for secondary treatment of wastewater. In addition, this method results in the creation of an eco-friendly and aesthetically pleasing zone, which, due to lower energy and material input, causes fewer negative environmental impacts [71,72].

Furthermore, successful integration of MFCs into CWs using low cost materials can further improve the overall economic viability, as the system shall be able to convert organic waste into green energy and harvest bioelectricity [73,74].

## 9. Conclusions

The Constructed Wetland Microbial Fuel Cell (CW-MFC) offers cutting edge technology for resolving the current issue of clean water shortage and ever-increasing energy needs. It accomplishes this by not only recovering water, but also producing bioelectricity from it. This makes CW-MFC a more affordable, environmentally friendly, and sustainable method of treating wastewater than conventional methods, which still require high energy and cost inputs, generate secondary pollutants, and are not environmentally friendly or aesthetically pleasing. CW-MFCs have been reported to reduce levels of recalcitrant contaminants such as dyes, xenobiotics, and heavy metals, in addition to removing COD, BOD, nitrogen, and phosphorous from the contaminated wastewater. One major limitation of this technology is its inability to produce enough power to be suitable for any practical application. Therefore, further focused and radical studies must be conducted in order to understand the dynamic ecosystem of CW-MFCs, so that they may be modified and fine-tuned in order to extract maximum energy outputs along with effective wastewater treatment. Moreover, studies on optimization of various parameters, including HRT, substrate feed, electrode materials and spacing, electrode microflora, macrophyte species, pollutant removal effectiveness, and

power generation, can be improved. A fundamental and scientific understanding of these factors will facilitate our understanding of the unnoticed features in this technology, which will allow for its successful implementation in the field for the treatment of wastewater and for harvesting bio-electricity.

**Author Contributions:** S.K. and D.P.: writing—original draft preparation; J.M. and N.P.: writing—reviewing and editing; R.R.: Conceptualization, writing, reviewing, editing, project administration and fund acquisition. All authors have read and agreed to the published version of the manuscript.

**Funding:** This research was funded by SERB, India via project sanction no. SPG/2021/004678 under SERB-DST, India.

**Institutional Review Board Statement:** Not relevant.

**Informed Consent Statement:** Not applicable.

**Data Availability Statement:** Not applicable.

**Acknowledgments:** The authors (R.R.) deeply acknowledge the financial support via project sanction no. SPG/2021/004678 under SERB-DST, India. R.R., J.M., S.K. and D.P. are thankful to Director MNNIT Allahabad for his support.

**Conflicts of Interest:** The authors declare no conflict of interest.

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
