# Peer review of "Constructed Wetland Coupled Microbial Fuel Cell: A Clean Technology for Sustainable Treatment of Wastewater and Bioelectricity Generation"

_fermentation, doi:10.3390/fermentation9010006_

Round 1
Reviewer 1 Report
This paper describes the use of CW-MFC for cost-effective, cleaner, and more sustainable approach for wastewater treatment. Results of the study may have important application in the field of wastewater treatment and energy recovery. Authors may wish to consider the following in revision of their manuscript.
1. Authors may wish to provide detailed information regarding characteristics data of raw wastewater for wetland treatment system.
2. Please provide mass balance for carbon for the wetland treatment system
3. Please provide data regarding how much electricity was generated fro the proposed wetland treatment system
4. Please provide cost data and cost benefit data regarding the proposed treatment system.
5. Please comment whether the proposed treatment system can meet effluent standards in author’s country.
6. Please provide design parameters of the proposed treatment systems.
Author Response
Responses to the Reviewers comments
Authors response to Reviewers comments on the manuscript "Constructed Wetland coupled Microbial Fuel Cell: A clean technology
for sustainable treatment of wastewater and bioelectricity generation" fermentation-1990962.
We appreciate the Editor’s and reviewers’ comments, suggestions, and time on our manuscript. These comments are all valuable and very helpful for revising and improving our manuscript. According to these comments and suggestions, we have carefully revised the manuscript which we hope to meet with approval. All the major corrections in the revised manuscript are highlighted in yellow.
The detailed responses to the comments are shown below and have been highlighted in yellow elsewhere needed:
S. No. |
Comments |
Response
|
|
Reviewer 1
|
|||
1. |
This paper describes the use of CW-MFC for cost-effective, cleaner, and more sustainable approach for wastewater treatment. Results of the study may have important application in the field of wastewater treatment and energy recovery. Authors may wish to consider the following in revision of their manuscript.
|
Authors are highly grateful to the reviewer for their words of appreciation and encouragement. |
|
2. |
1. Authors may wish to provide detailed information regarding characteristics data of raw wastewater for wetland treatment system. |
Authors are thankful for your important suggestion. Normally, the waste water affluent following primary treatment is further subjected to secondary treatment by CW, followed by tertiary treatment. (Fig 1.). The characteristics of waste water subjected to treatment in CW can have a range of values for COD, TN, TP, TSS, TDS, etc which shall differ according to the type of waste water (domestic sewage, grey-water, textile industry effluent, tannery effluent, food and beverage industry effluent), thus details of the same are not included. Also, the main focus of the manuscript is the integration of MFC in CW, and not just CW, emphasizing on CW shall dilute the main focus of the manuscript. |
|
3. |
Please provide mass balance for carbon for the wetland treatment system |
Authors are thankful for your valuable input. Necessary corrections have been made in the revised manuscript and carbon mass balance of a CW-MFC system has been included in Section 6.3.4., Page no. 12 |
|
4. |
Please provide data regarding how much electricity was generated from the proposed wetland treatment system
|
Authors have compiled the power output data (maximum power density and maximum current density) of various CW-MFC setups as reported by different workers in Table 2 and same has been discussed in section 7, pages 14 and 15 |
|
5. |
Please provide cost data and cost benefit data regarding the proposed treatment system. |
Thankyou for your valuable comment, the authors have included a section on economic considerations (section 8, page 15) which includes the cost analysis of CW-MFC. |
|
6. |
Please comment whether the proposed treatment system can meet effluent standards in author’s country. 6. . |
Thankyou for the very important query comment. According to the available literature, Constructed Wetlands have been reported to achieve effluent standards for small to medium scale domestic sewage and greywater, however special design and other considerations (use of specific plants and microbes) have to be taken care of, if toxic pollutants have to be targeted (Singh et al. 2021). Some of the literature regarding efficacy of CWs in pollutant removal and meet the effluent discharge standards in India are as follows: · Singh G, Gupta D, Singh G, Mishra VK (2021) Performance of horizontal flow constructed wetland for secondary treatment of domestic wastewater in a remote tribal area of Central India. Sustainable Environment Research, 31: 13 · https://www.cseindia.org/constructed-wetland-for-wastewater-treatment-at-indian-agriculture-research-institute-pusa-new-delhi-5632 · https://www.cseindia.org/constructed-wetland-to-treat-wastewater-at-indian-institute-of-technology-powai-mumbai-6209 o https://www.neeri.res.in/contents/banner_details/phytorid-wastewater-treatment-technology/5b193f6eaacd7#googtrans(en|en) |
|
7. |
Please provide design parameters of the proposed treatment systems. |
Thank you for your appreciation |
|
Thank-you once again for your time and consideration
Regards
Radha
Dr. Radha Rani
Department of Biotechnology,
Motilal Nehru National Institute of Technology Allahabad, Teliyarganj,
Prayagraj - 211004, Uttar Pradesh, India.
Tel: +91-532-2271240; Fax: +91-532-2271200
E-mail: radharani@mnnit.ac.in; raadharaani1982@gmail.com

Reviewer 2 Report
Hello authors,
I have read your review manuscript and I am satisfied with the written work overall.
Here are a few comments and suggestions that may improve the content and quality of this paper:
1. Abstract:
I feel that it needs to be refined to show that it is an abstract rather than a summary. No picture/figure needed to be added in this section. It should also address what this review paper really entails.
2. Line 37: where nearly 48%
3. Line 55: carbon here is it biomass , solid waste, or perhaps to cite and refer to which section/s in this paper the reader ought to be referring to.
4. Line 437, sec 6.3.4: Font larger for a number of lines here.
5. Perhaps to also show a more schematic technical -engineering diagram of the CW-MFC, some real photos of the experiment perhaps, instead of arrows representing electrods.
English wise is good. Perhaps to tidy up a little bit.
Thank you.
Author Response
Responses to the Reviewers comments
Authors response to Reviewers comments on the manuscript "Constructed Wetland coupled Microbial Fuel Cell: A clean technology
for sustainable treatment of wastewater and bioelectricity generation" fermentation-1990962.
We appreciate the Editor’s and reviewers’ comments, suggestions, and time on our manuscript. These comments are all valuable and very helpful for revising and improving our manuscript. According to these comments and suggestions, we have carefully revised the manuscript which we hope to meet with approval. All the major corrections in the revised manuscript are highlighted in yellow.
The detailed responses to the comments are shown below and have been highlighted in yellow elsewhere needed:
S. No. |
Comments |
Response
|
|
Reviewer 2
|
|||
1. |
I have read your review manuscript and I am satisfied with the written work overall. |
Thank you for your appreciation and important comments. The abstract has been revised according to your valuable suggestions. |
|
2. |
Line 37: where nearly 48%
|
Thankyou for the remark. The sentence has been reframed to make it grammatically correct. (Page 2) |
|
3. |
Line 55: carbon here is it biomass, solid waste, or perhaps to cite and refer to which section/s in this paper the reader ought to be referring to. |
Necessary correction has been incorporated in the revised manuscript. The statement was out of place in the paragraph, this it has been deleted. |
|
4. |
Line 437, sec 6.3.4: Font larger for a number of lines here. |
Necessary correction has been incorporated in the revised manuscript. |
|
5. |
Perhaps to also show a more schematic technical -engineering diagram of the CW-MFC, some real photos of the experiment perhaps, instead of arrows representing electrodes. |
In figure 6, which is schematic representation of a CW-MFC, electrodes are indicated by black dots, and the arrows indicate flow regime (vertical in this case). Authors agree to reviewers that some real site photos would add more value to the review, but as CW-MFC is still a budding technology and is not yet applied in field, getting site images is difficult. |
|
6. |
English wise is good. Perhaps to tidy up a little bit. |
Thankyou for your valuable suggestion. Authors have taken care to improve the grammar and language use in the revised version. |
|
Thank-you once again for your time and consideration
Regards
Radha
Dr. Radha Rani
Department of Biotechnology,
Motilal Nehru National Institute of Technology Allahabad, Teliyarganj,
Prayagraj - 211004, Uttar Pradesh, India.
Tel: +91-532-2271240; Fax: +91-532-2271200
E-mail: radharani@mnnit.ac.in; raadharaani1982@gmail.com

Reviewer 3 Report
1- The title is too long and needs to be shortened
2- All the given images need to be more clear and the resolution must increase
3- The author is using mixed font sizes and he needs to make only one font with one size
4- The conclusion section needs more explanation and the author have to mention the key challenges of this study and its merits as well.
5- The whole manuscript needs a language check by a native English speaker.
Author Response
Responses to the Reviewers comments
Authors response to Reviewers comments on the manuscript "Constructed Wetland coupled Microbial Fuel Cell: A clean technology
for sustainable treatment of wastewater and bioelectricity generation" fermentation-1990962.
We appreciate the Editor’s and reviewers’ comments, suggestions, and time on our manuscript.
These comments are all valuable and very helpful for revising and improving our manuscript. According to these comments and suggestions, we have carefully revised the manuscript which we hope to meet with approval. All the major corrections in the revised manuscript are highlighted in yellow.
The detailed responses to the comments are shown below and have been highlighted in yellow elsewhere needed:
S. No. |
Comments |
Response
|
|
|
Reviewer 3
|
||
1. |
The title is too long and needs to be shortened
|
Authors are thankful for your remarks. Title of the manuscript "Constructed Wetland coupled Microbial Fuel Cell: A clean technology |
|
2. |
All the given images need to be more clear and the resolution must increase
|
Authors are thankful for your valuation input and as suggested resolution of all the images have been improved in the revised version of the manuscript. |
|
3. |
The author is using mixed font sizes and he needs to make only one font with one size
|
Thank you for your valuable suggestion. Authors have made necessary corrections wrt use of fonts in the revised manuscript. |
|
4. |
The conclusion section needs more explanation and the author have to mention the key challenges of this study and its merits as well.
|
Thank you for your valuable suggestion. Authors have revised the conclusion section to make it more explanatory and cohesive in the revised manuscript. |
|
5. |
The whole manuscript needs a language check by a native English speaker.
|
The manuscript has been checked for English language use and grammar and care has been taken to avoid any such errors. |
|
Thank-you once again for your time and consideration
Regards
Radha
Dr. Radha Rani
Department of Biotechnology,
Motilal Nehru National Institute of Technology Allahabad, Teliyarganj,
Prayagraj - 211004, Uttar Pradesh, India.
Tel: +91-532-2271240; Fax: +91-532-2271200
E-mail: radharani@mnnit.ac.in; raadharaani1982@gmail.com

Round 2
Reviewer 1 Report
Authors have revised manuscript per suggestion sof reviewer. Manuscript is now acceptable for publication.